# REPARAMETERIZING 4DVAR WITH NEURAL FIELDS

## ABSTRACT

Four-dimensional variational data assimilation (4DVAR) is a cornerstone of numerical weather prediction, but its cost function is difficult to optimize and computationally intensive. We propose a neural field-based reformulation in which the full spatiotemporal state is represented as a continuous function parameterized by a neural network. This reparameterization removes the time-sequential dependency of classical 4DVAR, enabling parallel-in-time optimization in parameter space. Physical constraints are incorporated directly through a physics-informed loss, simplifying implementation and reducing computational cost. We evaluate the method on the two-dimensional incompressible Navier–Stokes equations with Kolmogorov forcing. Compared to a baseline 4DVAR implementation, the neural reparameterized variants produce more stable initial condition estimates without spurious oscillations. Notably, unlike most machine learning-based approaches, our framework does not require access to ground-truth states or reanalysis data, broadening its applicability to settings with limited reference information.

## 1 INTRODUCTION

Since Richardson's pioneering work on rational weather forecasting, numerical weather prediction (NWP) has become a cornerstone of modern society (Lynch, 2008). NWP is typically posed as an initial value problem,[1]

$$\frac{\partial u}{\partial t} = \mathcal{F}(u), \qquad (t, x) \in (0, T) \times \Omega, \tag{1a}$$

$$u(0, x) = u_0(x), \qquad x \in \Omega, \tag{1b}$$

where $\Omega \subset \mathbb{R}^d$ is open and bounded, $u : [0, T] \times \Omega \to \mathbb{R}^N$, and $\mathcal{F}$ is a differential operator governing the dynamics. In practice, however, the initial condition $u_0$ is not fully known—satellite and ground-based measurements are both noisy and spatially sparse. While interpolation methods exist (Daley, 1993, Chapter 4), the chaotic nature of atmospheric dynamics (Lorenz, 1963; 1993) causes even small errors to amplify rapidly, degrading forecast accuracy. Thus, accurate estimation of the initial state is a central challenge in NWP.

Data assimilation tackles this challenge by combining models with observations to produce improved estimates of physical states. Among the many approaches (Evensen, 1994; Hunt et al., 2007; Liu et al., 2008), we focus on the four-dimensional variational method (4DVAR) (Le Dimet & Talagrand, 1986; Rabier et al., 1998). The name highlights its formulation in both space (three dimensions) and time (the fourth dimension), making it particularly well-suited for atmospheric and geophysical applications.

### 1.1 FOUR-DIMENSIONAL VARIATIONAL DATA ASSIMILATION

The 4DVAR is usually formulated on a discrete state-space model,

$$u_{k+1} = F(u_k), \tag{2a}$$

$$y_k = H(u_k) + \varepsilon_k, \tag{2b}$$

where $u_k$ denotes the system state at time $t_k$, $y_k$ is the corresponding observation, $F : \mathbb{R}^{d_u} \to \mathbb{R}^{d_u}$ represents the dynamical model, and $H : \mathbb{R}^{d_u} \to \mathbb{R}^{d_y}$ is the observation operator. The term

---

[1]Often formulated using the primitive equations (Bjerknes et al., 2009; Lions et al., 1992).

$\varepsilon_k$ accounts for observation noise. This discrete model arises from approximating the continuous dynamics in eq. (1). Specifically, the domain $\Omega$ is discretized into a spatial grid, the state $u$ is represented by its values on this grid, and the states evolve forward in time from $u_0$ following the model eq. (1), which encodes physical principles. For convenience, we use $u$ to denote both the exact continuous solution and its numerical approximation when the meaning is clear from context. With these discretizations in place and a specification of how observations are generated, the continuous formulation in eq. (1) reduces to the state-space system in eq. (2). In practice, the dimension of the state space is typically much larger than that of the observation space ($d_u \gg d_y$), which underscores the difficulty of accurately reconstructing the full system state from sparse and noisy data.

The classical 4DVAR approach estimates the initial condition $u_0^\star$ by minimizing the cost function

$$J(u_0) = \|u_0 - u_b\|_B^2 + \sum_{k=1}^{K} \|H(u_k) - y_k\|_{R_k}^2 \,, \tag{3}$$

where $u_b$ is a background (prior) state—usually the most recent forecast, and $B$ and $R_k$ denote the error covariance matrices of the background and the observations, respectively. Misfits are measured in Mahalanobis distance, defined as $\|x\|_A^2 = (A^{-1}x, x)$, with $(\cdot, \cdot)$ the standard Euclidean inner product.

Intuitively, incorporating future observations $\{y_k\}_{k=1}^{K}$ should improve the estimate of the initial condition. In practice, however, this benefit is realized only if the cost function eq. (3) can be minimized effectively. The minimization itself poses significant challenges, which we outline next.

**Non-uniqueness.** The nonlinearity of eq. (2a) makes the cost function eq. (3) non-convex. Unlike convex problems, where global minimizers are uniquely defined and reliably attainable, non-convex problems admit multiple local minima (Nocedal & Wright, 1999). Moreover, in dissipative systems, small perturbations to the initial condition often decay quickly, which means that very different initial states can evolve into trajectories that are equally consistent with the same set of observations. This non-uniqueness complicates the optimization landscape, making convergence dependent on the initial guess and raising interpretability challenges for the estimated solution.

**Computational cost.** Minimizing the 4DVAR cost function typically relies on iterative gradient-based methods. However, the sequential nature of eq. (2a) makes gradient evaluation expensive. The adjoint method (Errico, 1997; Johnson, 2012) computes gradients with a runtime comparable to the forward model, but it requires access to all intermediate states, leading to prohibitively high memory costs. Checkpointing strategies (Griewank, 1992; Griewank & Walther, 2000) alleviate this by recomputing selected states, reducing memory usage at the expense of additional runtime and careful tuning. While parallel-in-time methods such as the Parareal algorithm (Lions et al., 2001) offer another possible remedy, their application to 4DVAR remains in its early stages (Bhatt et al., 2025).

## 1.2 CONTRIBUTIONS

The challenges discussed above motivate alternative formulations of 4DVAR that retain its strengths while reducing computational and implementation burdens.

Practical implementations of 4DVAR often rely on hierarchical refinement strategies (Bonavita et al., 2018). For example, inner loops solve quadratic approximations using tangent-linear models and adjoint methods, while outer loops progressively refine the solution from coarse to fine spatial resolutions.[2] This design implicitly prioritizes large-scale corrections before resolving finer details, thereby reducing the risk of instability (Veersé & Thépaut, 1998).

Interestingly, neural fields exhibit an analogous property known as *spectral bias*. They naturally capture smooth, large-scale structures early in training and only later fit finer details (Rahaman et al., 2019). This parallel suggests that neural reparameterization can inherit the same stabilizing effect that classical 4DVAR achieves through carefully engineered hierarchical refinements. Motivated

---

[2]ERA5 uses 3 inner loop resolutions: TL95, TL159, and TL255 (Hersbach et al., 2020, Table 2).

$$\left\| H\left(\boxed{u_0} \xrightarrow{F^{(K)}} \boxed{u_K}\right) - \boxed{y_K} \right\| \quad \left\| H\left(\theta \mapsto \boxed{u_0} \xrightarrow{F^{(K)}} \boxed{u_K}\right) - \boxed{y_K} \right\| \quad \left\| H\left(\theta \mapsto \boxed{u_K}\right) - \boxed{y_K} \right\| + L_{\text{PINN}}$$

(a) VANILLA-4DVAR          (b) NEURAL-4DVAR          (c) PINN-4DVAR

Figure 1: Schematics of three 4DVAR cost functions. (a) VANILLA-4DVAR estimates the initial condition by minimizing observation misfits. (b) NEURAL-4DVAR reparameterizes the initial condition with a neural network. (c) PINN-4DVAR parameterizes the full spatiotemporal state $u^\theta(t, x)$ and enforces governing dynamics through a physics-informed loss $L_{\text{PINN}}$.

by this connection, we propose to reformulate 4DVAR by representing spatiotemporal states with neural fields.[3]

We explore two strategies for performing 4DVAR with neural fields. The first strategy parameterizes only the initial condition with a neural field, leveraging spectral bias to stabilize the optimization process. We examine this effect in detail through an energy spectrum analysis in Section 4.1. The second strategy parameterizes the entire spatiotemporal state with a neural field, incorporating physical constraints via physics-informed losses (Raissi et al., 2019). This formulation eliminates the time-sequential dependence of classical 4DVAR and enables parallel-in-time optimization, thereby achieving a **33%** reduction in runtime.

Unlike most machine learning-based data assimilation methods, neural parameterized variants operate without requiring access to ground-truth states or reanalysis products such as ERA5 (Hersbach et al., 2020). This independence from reference data makes the approach particularly attractive in domains where observations are limited or incomplete, and points toward scalable data assimilation in high-dimensional systems.

## 2 METHOD

We build on the idea of neural reparameterization (Hoyer et al., 2019), in which classical state variables are replaced with neural networks. In our setting, we adopt neural fields (Sitzmann et al., 2020), which represent a signal as a continuous mapping $\mathbf{x} \mapsto u^\theta(\mathbf{x})$. Neural fields have proven effective in diverse domains ranging from novel-view synthesis (Mildenhall et al., 2021; Li et al., 2025) to video representation (Chen et al., 2021; Kwan et al., 2023; Wu et al., 2024). Their mesh-free nature makes them particularly appealing for data assimilation, where observations may be irregular or sparse.

**Simplified 4DVAR cost function.** Throughout this work, we adopt a simplified version of the 4DVAR cost function,

$$J_{\text{Vanilla}}(u_0) = \sum_{k=0}^{K} \| H(u_k) - y_k \|^2. \tag{4}$$

This form omits background and covariance terms, allowing us to isolate the effects of neural reparameterization without additional statistical or modeling assumptions. We refer to this baseline implementation as VANILLA-4DVAR.

### 2.1 PARAMETERIZING THE INITIAL CONDITION

In the first strategy, the initial condition $u_0(x)$ is represented by a neural field $u_0^\theta(x)$, with parameters $\theta$. The 4DVAR cost function eq. (4) is then reparameterized as

$$J_{\text{Neural}}(\theta) = J_{\text{Vanilla}}(u_0^\theta). \tag{5}$$

---

[3]In the context of neural fields, spectral bias is often considered a limitation (Tancik et al., 2020; Sitzmann et al., 2020; Kang et al., 2025), as it hinders representation of fine-scale details. Yet this bias can also be leveraged—Zhang et al. (2022), for example, combined neural networks with classical partial differential equation (PDE) solvers, letting neural networks capture large-scale structures while iterative solvers refine finer details.

For notational simplicity, we denote the observation misfit by $J$. The optimization is now performed over $\theta$, rather than directly over $u_0$. We refer to this formulation as NEURAL-4DVAR.

Although no formal guarantees exist that optimization in parameter space is easier, empirical evidence from related domains (Hoyer et al., 2019; Krueger & Ward, 2025) suggests improved convergence and solution quality. The trade-off is increased per-iteration cost, since evaluating $u^\theta$ requires a forward pass through the network.[4]

## 2.2 PARAMETERIZING THE ENTIRE STATE

The second strategy parameterizes the full spatiotemporal state $u(t, x)$ with a neural field $u^\theta(t, x)$, defined for $t \in [0, T]$ and $x \in \Omega$.

Physical constraints are incorporated through physics-informed losses (Raissi et al., 2019). Recall that a candidate solution to eq. (1) can be obtained by minimizing

$$L(\theta) = \underbrace{\int_{(0,T)\times\Omega} \Big(\frac{\partial u^\theta}{\partial t} - \mathcal{F}(u^\theta)\Big)^2 \mathrm{d}t\mathrm{d}x}_{L_{\mathrm{PINN}}(\theta)\ \text{(physics constraint)}} + \underbrace{\int_\Omega \big(u^\theta(0, x) - u_0(x)\big)^2 \mathrm{d}x}_{\text{initial condition misfit}}. \tag{6}$$

To estimate the initial condition $u_0$ from observations, we modify this formulation to a composite loss:

$$J_{\mathrm{PINN}}(\theta) = L_{\mathrm{PINN}}(\theta) + \lambda_{\mathrm{data}} \underbrace{\sum_{k=0}^{K} \big(H(u_k^\theta) - y_k\big)^2}_{\text{observation misfit}}, \tag{7}$$

where $\lambda_{\mathrm{data}}$ balances physical consistency and data fit. Minimizing eq. (7) defines the PINN-4DVAR algorithm.

This formulation offers a key scalability advantage. Unlike VANILLA- and NEURAL-4DVAR, evaluating $J_{\mathrm{PINN}}(\theta)$ does not require spatial discretization followed by sequential time integration. Instead, observation misfits can be computed independently across time, enabling parallel-in-time optimization. Empirical comparisons of computational cost are reported in Table 2.

For clarity, Figure 1 illustrates the three variants: VANILLA-, NEURAL-, and PINN-4DVAR.

**Remark.** Since the early work of Lagaris et al. (1998); Raissi et al. (2019), PINNs have faced challenges in solving PDEs reliably (Krishnapriyan et al., 2021), even though convergence guarantees exist in theory (Jo et al., 2020; Shin et al., 2020). Issues are often attributed to stiff optimization dynamics (Wang et al., 2021; De Ryck et al., 2024; Lee et al., 2025) or violations of temporal causality (Wang et al., 2024). In the data assimilation setting, however, the presence of observations mitigates these issues—observational constraints stabilize training and reduce temporal causality violations.

**Two-step optimization: Hybrid-4DVAR.** Finally, we consider a two-step strategy. We first apply PINN-4DVAR to exploit its parallel efficiency, then refine the result with NEURAL-4DVAR. Because PINN-4DVAR provides an effective initialization, the additional overhead of this preliminary step is modest, while the overall gain in accuracy is substantial. We refer to this combined approach as HYBRID-4DVAR.

## 3 NUMERICAL RESULTS

**Kolmogorov flow.** To evaluate the proposed method, we consider the two-dimensional incompressible Navier–Stokes equations with Kolmogorov forcing, commonly referred to as Kolmogorov flow (Boffetta & Ecke, 2012; Chandler & Kerswell, 2013). Following the setup of Frerix et al.

---

[4]In practice, faster convergence may outweigh this additional overhead.

| SPARSITY | $2^2$ (25%) | $4^2$ (6.25%) | $8^2$ (1.56%) | $16^2$ (0.39%) | $32^2$ (0.1%) |
|---|---|---|---|---|---|
| INTERP | 8.87E-3 | 5.31E-2 | 2.70E-1 | 8.44E-1 | 1.18E0 |
| VANILLA | **5.08E-3** | 3.76E-2 | 2.15E-1 | 9.27E-1 | 1.68E0 |
| NEURAL | 8.87E-3 | 2.52E-2 | 7.80E-2 | 1.56E-1 | 5.37E-1 |
| PINN | 8.53E-2 | 8.54E-2 | 8.74E-2 | 1.56E-1 | 3.62E-1 |
| HYBRID | 1.95E-2 | **1.97E-2** | **2.15E-2** | **3.62E-2** | **1.92E-1** |

Table 1: Relative $L^1$ errors between estimated and true initial conditions under varying levels of observation sparsity. Boldface and underline indicate the best and the second best in each column. As expected, the accuracy of two baselines degrades as observations become sparser. By contrast, PINN-4DVAR achieves competitive accuracy at high sparsity. NEURAL- and HYBRID-4DVAR maintain better performance overall.

(2021), the governing equations are

$$\frac{\partial \mathbf{u}}{\partial t} + (\mathbf{u} \cdot \nabla)\mathbf{u} + \frac{1}{\rho}\nabla p = \nu\nabla^2\mathbf{u} - 0.1\mathbf{u} + \sin(4y)\hat{\mathbf{x}}, \qquad (t, x, y) \in (0, T) \times [0, 2\pi]^2, \quad \text{(8a)}$$

$$\nabla \cdot \mathbf{u} = 0\,. \tag{8b}$$

We set the density $\rho = 1$, viscosity $\nu = 10^{-2}$, $\hat{\mathbf{x}} = [1, 0]^T$, $T = 0.5$, and impose periodic boundary conditions. The velocity field $\mathbf{u}$ has two components (in the $x$ and $y$ directions). The incompressibility condition eq. (8b) allows a reformulation in terms of vorticity. Taking the curl of eq. (8a) yields

$$\frac{\partial \omega}{\partial t} + (\mathbf{u} \cdot \nabla)\omega = \nu\nabla^2\omega - 0.1\omega - 4\cos(4y). \tag{9}$$

For detailed derivation, we refer the reader to Appendix A.2. For numerical solutions, we adopt a Fourier collocation method for spatial discretization and an implicit–explicit (IMEX) scheme for time integration of the semi-discretized system. Further implementation details are provided in Appendix A.3.

**Simulated observations.** To generate synthetic observations, we first integrate eq. (9) from a random initial condition up to $t = 10$, which we then take as the initial condition. From this state, we integrate forward to $t = 0.5$, recording velocity snapshots at $t \in \{0, 0.05, \ldots, 0.5\}$. Observations are produced by applying a subsampling operator—for sparsity level $k^2$, the operator selects every $(kn+1)$-th component in each spatial direction, so that the sparsity reflects subsampling along both $x$ and $y$. For VANILLA- and NEURAL-4DVAR, the initial condition at $t = 0$ is obtained by bicubic interpolation of the observed data.

**Setup.** All neural parameterizations use the separable physics-informed neural network (SPINN) architecture (Cho et al., 2024)—Appendix A.5 provides further description. For VANILLA-4DVAR, we employ the L-BFGS optimizer (Liu & Nocedal, 1989), while NEURAL- and PINN-4DVAR use the AdamW optimizer (Loshchilov & Hutter, 2019). Additional hyperparameter settings are provided in Appendix A.6.

### 3.1 ACCURACY TESTS

**Sparse observation.** We first investigate the effect of observation sparsity on the accuracy of the estimated initial states. The sparsity level is varied from $2^2$ to $32^2$, corresponding to approximately 25% down to 0.1% of the full state being observed. Table 1 reports the relative $L^1$ errors between the ground-truth initial vorticity and the estimates produced by different methods. As an additional baseline, we include INTERP, which applies bicubic interpolation to the observations at $t = 0$. As expected, accuracy deteriorates as observations become sparser. Nevertheless, the neural reparameterized variants consistently achieve lower errors than both baselines, demonstrating improved robustness under limited observational coverage.[5]

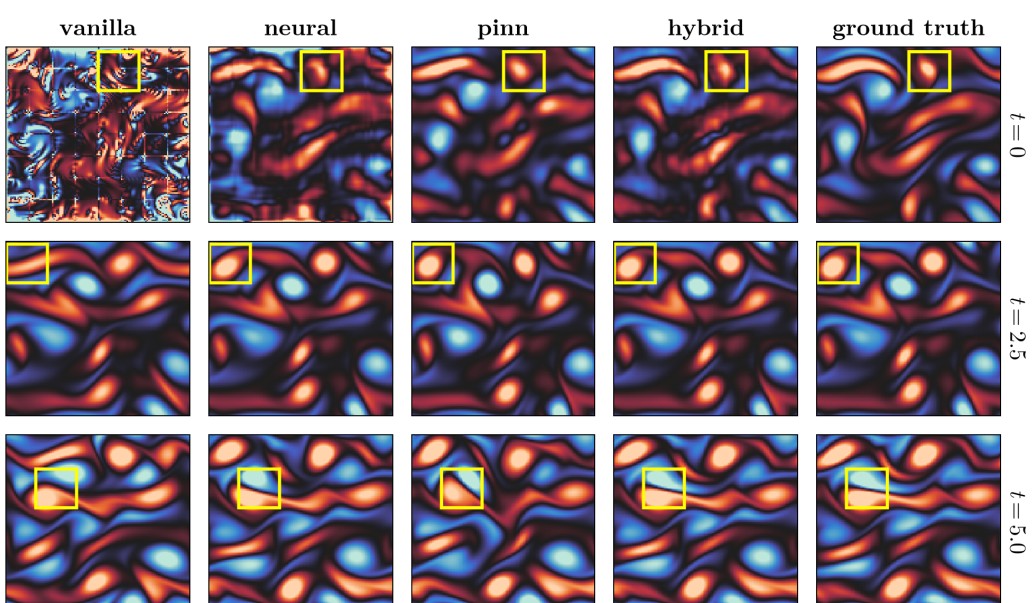

Figure 2: Rollout test results from assimilated initial conditions. Rows correspond to time snap-shots at $t \in \{0, 2.5, 5\}$. The first four columns show forecasts obtained using different assimilation methods, and the last column shows the ground-truth states. The VANILLA-4DVAR initialization exhibits a spurious high-frequency perturbation that decays quickly during the forecast, whereas neural reparameterized variants produce smoother and more stable rollouts. This phenomenon is further examined in the energy spectrum analysis of Section 4.1.

**Rollout test.** Recall that observations were assimilated over the time interval $[0, 0.5]$. In many applications, however, the central objective is the quality of forecasts generated from the assimilated initial condition. To assess this, we roll out the estimated initial conditions with the numerical solver up to $T = 5$.

Figure 2 shows the resulting forecasts. The PINN-4DVAR solution preserves key vorticity features of the ground-truth state up to $t = 2.5$, while the hybrid method achieves the best over-all fidelity across the horizon. Remarkably, even the simple neural reparameterization of the initial condition—without the full physics-informed framework—achieves strong perfor-mance, underscoring the value of neural param-eterization for improving forecast accuracy.

Figure 3 quantifies forecast accuracy through the growth of relative $L^1$ errors over time. Errors initially decrease, reflecting the dis-sipative nature of the benchmark problem, where drag and viscosity quickly damp small perturbations (Frerix et al., 2021). While PINN-4DVAR achieves slightly lower assim-ilation error than NEURAL-4DVAR (3.62E-1 vs. 5.37E-1), the forecasts from NEURAL-4DVAR remain more accurate at longer hori-zons. This behavior is expected—NEURAL-4DVAR is tailored to the same numerical

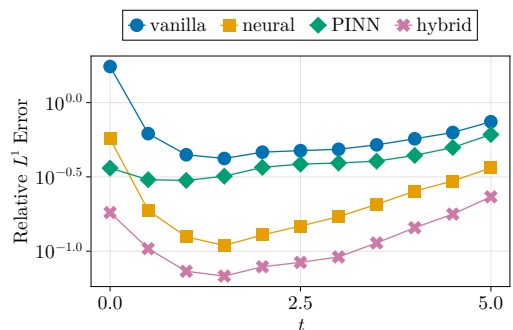

Figure 3: Relative $L^1$ errors during the rollout test. While NEURAL-4DVAR starts from a less accurate initial condition than PINN-4DVAR, its errors decay rapidly once forecasting begins, yielding a more accurate long-term prediction at the final time. This reflects the alignment between NEURAL-4DVAR and the numerical solver used for forecasting.

---

[5] For validation of our baseline implementation, see Appendix A.4.

| COMPUTATIONAL COST | time (s) | memory (MB) |
|---|---|---|
| VANILLA ($10^3$ steps, w/o checkpointing) | 351 | 17102 |
| VANILLA ($10^3$ steps, w/ checkpointing) | 455 | 1742 |
| NEURAL ($10^3$ steps, w/ checkpointing) | 420 | **1290** |
| PINN ($10^4$ steps) | **52** | 1786 |
| HYBRID ($10^4$ PINN + 500 NEURAL steps) | 283 | 1786 |

Table 2: Wall-clock time and memory usage for optimization. Results highlight the comparable memory requirements and the significant runtime speedup of PINN-4DVAR enabled by its parallel-in-time formulation.

solver used for forecasting, giving it a structural advantage in rollout accuracy. By contrast, PINN-4DVAR enforces governing dynamics through a physics-informed loss. While this may yield slightly less accurate short-term forecasts, it offers the potential for better robustness when the forecasting model differs from the assimilation model or when observations are sparse or noisy.

## 3.2 COMPUTATIONAL EFFICIENCY

Table 2 reports wall-clock time and memory usage for VANILLA-, NEURAL-, and PINN-4DVAR for Section 3.1. Despite parameterizing the full spatiotemporal state, the memory requirements of PINN-4DVAR remain comparable to those of VANILLA-4DVAR.[6] Further reductions may be possible using neural compression techniques (Dupont et al., 2021).

The most notable difference lies in the runtime. VANILLA- and NEURAL-4DVAR require sequential forward and backward passes through time, creating a strong bottleneck that limits scalability, especially on modern parallel hardware. By contrast, the parallel-in-time formulation of PINN-4DVAR yields an approximately **8×** speedup in overall runtime.[7] While the advantage of SPINN may diminish for irregular observation grids, enforcing physical constraints via $L_{\text{PINN}}(\theta)$ remains computationally inexpensive so long as the domain retains a tensor-product structure.

## 4 ABLATION STUDY

To disentangle the effects of neural reparameterization and the physics-informed loss, we conduct two ablation studies, which we outline below.

### 4.1 EFFECT OF NEURAL REPARAMETERIZATION

We begin by evaluating the impact of neural reparameterization through a comparison of two cost functions: the classical formulation $J_{\text{Vanilla}}$ and its neural counterpart $J_{\text{Neural}}$. Both quantify observation misfits, but they operate over different domains: the former on the physical state $u_0$, and the latter on the neural network parameters $\theta$.

**Cost reduction vs. state accuracy.** Panel (**A**) of Figure 4 shows that both classical and neural formulations consistently decrease their respective cost functions over the course of optimization. However, the relative $L^1$ errors in panel (**B**) reveal an important distinction: VANILLA-4DVAR reduces the cost but fails to improve the accuracy of the estimated initial condition, whereas NEURAL-4DVAR reduces both the cost and the error, leading to a more faithful reconstruction.[8]

**Energy spectrum analysis.** Panel (**C**) of Figure 4 compares the energy spectra of assimilated and ground-truth initial velocities. In NEURAL-4DVAR, the spectrum remains stable during optimization, whereas VANILLA-4DVAR injects energy into the high-wavenumber regime, producing the

---

[6]The slightly higher memory footprint of VANILLA-4DVAR compared to NEURAL-4DVAR arises from the L-BFGS optimizer, which stores a history of 10 gradients.

[7]If we exclude the just-in-time compilation time, then the speedup increases more.

[8]Still, minimizing $J_{\text{Neural}}$ does not always translate into improved state accuracy, as illustrated in Table 4.

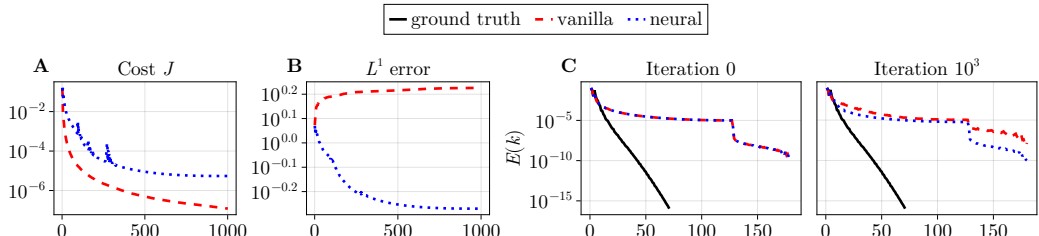

Figure 4: Comparison of VANILLA-4DVAR and NEURAL-4DVAR. (**A**) Cost function decays over optimization steps. (**B**) Relative $L^1$ error between assimilated and ground-truth initial conditions. For VANILLA-4DVAR (red, dashed), reducing the observation misfit does not necessarily improve state accuracy. In contrast, NEURAL-4DVAR (blue, dotted) achieves simultaneous cost and error reduction. (**C**) Energy spectra at the 0-th and $10^3$-rd iterations, showing growth of high-wavenumber energy in VANILLA-4DVAR but stability under NEURAL-4DVAR.

| NOISE ($\sigma$) | 5% | 10% | 15% | 20% | 25% |
|---|---|---|---|---|---|
| INTERP | 1.18E0 | 1.17E0 | 1.18E0 | 1.17E0 | 1.20E0 |
| VANILLA | 1.58E1 | 2.46E1 | 1.05E1 | 2.46E1 | 3.36E1 |
| NEURAL | 1.91E0 | 5.32E0 | 1.05E1 | 5.32E0 | 1.64E1 |
| PINN | **3.82E-1** | **4.08E-1** | **4.44E-1** | **4.74E-1** | **4.97E-1** |
| HYBRID | 2.12E0 | 5.43E0 | 8.10E0 | 1.06E1 | 1.28E1 |

Table 3: Relative $L^1$ errors between estimated and true initial conditions under varying levels of observation noise. Boldface indicates the best in each column. PINN-4DVAR consistently achieves the lowest error across all noise levels, indicating robustness of the physics-informed formulation.

artifacts visible in Figure 2 (first column). The spectral dynamics of PINN-4DVAR and HYBRID-4DVAR are shown in Figures 9 and 10.

This difference reflects the spectral bias of neural fields (Rahaman et al., 2019), which naturally emphasize smooth, large-scale structures before fitting high-frequency details. Although often considered a limitation for fine-scale recovery, in data assimilation this bias acts as an implicit regularizer, suppressing high-frequency artifacts and stabilizing optimization.

### 4.2 EFFECT OF THE PHYSICS-INFORMED LOSS

**Robustness under noisy observations.** Observations from the real world contain noise. Here, we compare the accuracy of estimated initial conditions from two baselines and neural reparameterized variants with the presence of data noise. For simplicity, we consider time-independent noise from the normal distribution, $N(0, \sigma^2 I)$. We add the random noise $\varepsilon$ after applying the subsampling operator of the sparsity level $32^2$ to prepare the observations.

Table 3 presents relative $L^1$ errors between estimated initial conditions and the true initial condition, with noisy observation data. Surprisingly, the estimated initial conditions from PINN-4DVAR show lower error levels across all noise levels. In particular, adding 5% noise to observations significantly deteriorates initial estimates from all methods, depending on numerical solver, suggesting the robustness of imposing physical constraints through the PINN loss.

**Regression baseline.** A natural question is whether the improved accuracy of PINN-4DVAR stems merely from the zero-shot super-resolution capability of neural fields (Shocher et al., 2018; Feng et al., 2024), rather than from the physics-informed loss. To investigate this, we compare PINN-4DVAR against a baseline that minimizes only the observation misfit,

$$J_{\text{Regression}}(\theta) = \sum_{k=0}^{K} \|H(u_k^\theta) - y_k\|^2,$$

omitting the physics-informed term $L_{\mathrm{PINN}}(\theta)$. As shown in Table 6, removing the physics-informed loss degrades accuracy in settings with sparse or noisy observations, highlighting the critical role of physical consistency in improving assimilation outcomes.

# 5 DISCUSSION

## 5.1 RELATED WORKS

**Machine learning and data assimilation.** A growing body of work explores ML methods for data assimilation. Notable examples include learned observation operators for 4DVAR (Frerix et al., 2021), and diffusion model-based approaches (Rozet & Louppe, 2023; Huang et al., 2024). Ensemble-free neural filters have also been proposed (Bocquet et al., 2024; Oh et al., 2025). In particular, Li et al. (2024) introduced data assimilation frameworks operating in latent spaces using coordinate-based MLPs. Recently, Yang et al. (2025) introduced Tensor-Var, which performs 4DVAR linearization in a kernel feature space and, notably, evaluated it on satellite observation data—a benchmark rarely used in ML-based data assimilation. However, these methods typically assume access to ground truth states or high-quality reanalysis datasets such as ERA5 (Hersbach et al., 2020), limiting direct comparison with our approach, which does not rely on such data. While Bao et al. (2024) presents a related score-based filtering method, their approach uses only historical observations, whereas ours incorporates future observations, rendering direct comparison challenging.

**Physics-informed neural networks for inverse problems.** PINNs have been widely explored for solving inverse problems in dynamical systems. Raissi et al. (2019) demonstrated their use in estimating unknown parameters in PDEs, such as the viscosity in Burgers' equation, while Raissi et al. (2020) applied PINNs to recover pressure fields from velocity data under the Navier–Stokes equations. Subsequent work has applied these methods to wider problems: for example, He et al. (2020) addressed subsurface transport by estimating states (hydraulic head and concentration) together with conductivity parameters in a steady-state setting. Son & Lee (2024) estimated parameters in thermoacoustic oscillators by transforming stochastic differential equations into a Fokker–Planck formulation with a likelihood-based loss, and Jo et al. (2024) used PINNs to infer transduction time distributions in delayed ODE models from final responses. To our knowledge, explicit connections between PINNs and classical 4DVAR have not yet been established, despite this progress.

## 5.2 LIMITATIONS AND FUTURE WORK

Our study focuses on an idealized setting, and extending the framework to more realistic applications of numerical weather prediction (NWP) remains future work. Key challenges include assimilating satellite data with non-local observation operators, handling three-dimensional globe-like geometries in general circulation models, and incorporating background error covariances with realistic noise models. Coupling with ocean and land components is another crucial step toward Earth-system data assimilation. In addition, comparisons with advanced 4DVAR implementations (Bannister, 2008) will be important for practical impact.

Another open direction concerns the treatment of physical constraints. PINN-4DVAR enforces dynamics weakly through PDE residuals, which suggests a connection to weak-constraint 4DVAR (Ngodock et al., 2017). Exploring this link, along with incorporating model uncertainty into the physics-informed loss, could strengthen robustness. Finally, advances in neural fields and PINNs—such as compact representations (Kerbl et al., 2023), and spectral loss (Du et al., 2024)—offer promising opportunities to further improve both accuracy and scalability.

## 5.3 CONCLUSION

In this work, we introduced a neural field-based reparameterization of 4DVAR and presented its effectiveness on the two-dimensional Kolmogorov flow. Our results show that neural parameterization, combined with physics-informed constraints, can improve both accuracy and efficiency over classical formulations. We believe neural reparameterizations of 4DVAR can form the basis of hybrid approaches combining classical 4DVAR rigor with ML flexibility in operational NWP.

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

## A EXPERIMENTAL DETAILS

This appendix provides the experimental details underlying Section 3.

**Software.** Python scientific computing ecosystem (Harris et al., 2020; Virtanen et al., 2020) and the JAX ecosystem (Bradbury et al., 2018; DeepMind et al., 2020; Blondel et al., 2022; Kochkov et al., 2021; Conlin, 2023) for numerical experiments. Makie.jl for visualization (Danisch & Krumbiegel, 2021).

### A.1 INCREMENTAL FORM

In numerical weather prediction, incremental formulations combined with background-error covariance preconditioning have proven effective (Trémolet, 2007). In our case, however, the background covariance matrix would require memory quadratic in the state dimension, which is prohibitive (e.g., $\sim 16$ GB for $u \in \mathbb{R}^{256 \times 256}$). Therefore, we shall consider only the following incremental formulation of the simplified 4DVAR cost function (4):

$$J(\delta u_0) = \sum_{k=0}^{K} \|y_k - H(u_k)\|^2,$$

where $u_0 = \hat{u}_0 + \delta u_0$. Here $\hat{u}_0$ is an initial guess. In our experiment, we interpolated the observations at $t = 0$ with the bicubic interpolation for $\hat{u}_0$. For VANILLA and NEURAL-4DVAR algorithms, we parameterized $\delta u_0$ with neural networks, instead of $u_0$.

### A.2 DERIVATION OF THE VORTICITY FORM

Starting from the incompressible Navier–Stokes equations with Kolmogorov forcing (eq. (8))

$$\frac{\partial \mathbf{u}}{\partial t} + (\mathbf{u} \cdot \nabla)\mathbf{u} + \frac{1}{\rho}\nabla p = \nu\nabla^2\mathbf{u} - 0.1\mathbf{u} + \sin(4y)\,\hat{\mathbf{x}},$$
$$\nabla \cdot \mathbf{u} = 0,$$

We take the two-dimensional curl of the momentum equation:

$$\nabla \times \left(\frac{\partial \mathbf{u}}{\partial t} + (\mathbf{u} \cdot \nabla)\mathbf{u} + \frac{1}{\rho}\nabla p\right) = \nabla \times \left(\nu\nabla^2\mathbf{u} - 0.1\mathbf{u} + \sin(4y)\,\hat{\mathbf{x}}\right).$$

The pressure gradient term vanishes after taking the curl, and introducing the vorticity $\omega = \nabla \times \mathbf{u}$ gives

$$\frac{\partial \omega}{\partial t} + \nabla \times \left[(\mathbf{u} \cdot \nabla)\mathbf{u}\right] = \nu\nabla^2\omega - 0.1\omega - 4\cos(4y).$$

To simplify the nonlinear term $(\mathbf{u} \cdot \nabla)\mathbf{u}$, we use the vector identity

$$(\mathbf{u} \cdot \nabla)\mathbf{u} = \nabla\left(\tfrac{1}{2}|\mathbf{u}|^2\right) - \mathbf{u} \times \omega.$$

Taking the curl yields

$$\nabla \times \left[(\mathbf{u} \cdot \nabla)\mathbf{u}\right] = \nabla \times \left[\nabla\left(\tfrac{1}{2}|\mathbf{u}|^2\right) - \mathbf{u} \times \omega\right]$$
$$= -\nabla \times (\mathbf{u} \times \omega),$$

since the curl of a gradient is zero. Expanding the remaining term,

$$-\nabla \times (\mathbf{u} \times \omega) = \omega(\nabla \cdot \mathbf{u}) - (\omega \cdot \nabla)\mathbf{u} + (\mathbf{u} \cdot \nabla)\omega.$$

By incompressibility, $\nabla \cdot \mathbf{u} = 0$. In two dimensions, $\omega$ is perpendicular to the flow plane, so $\omega \cdot \nabla = 0$. Hence, the nonlinear term reduces to $(\mathbf{u} \cdot \nabla)\omega$.

Collecting terms, the vorticity equation becomes

$$\frac{\partial \omega}{\partial t} + (\mathbf{u} \cdot \nabla)\omega = \nu\nabla^2\omega - 0.1\omega - 4\cos(4y).$$

### A.3 NUMERICAL SOLVER

The numerical method was implemented based on JAX-CFD (Kochkov et al., 2021).

**Spatial discretization.** Spatial discretization was performed with the Fourier collocation method with $N_x = 256$ along each direction as implemented in JAX-CFD. In vorticity form (eq. (9)), velocity was recovered from vorticity by solving $-\Delta\psi = \omega$, with $\mathbf{u} = (\partial_y\psi, -\partial_x\psi)$ following Yin et al. (2004). This step is straightforward in Fourier space, since the Laplacian is diagonal.

**Temporal integration.** Time integration employed a 2N storage Runge–Kutta scheme of order (5,4) (Carpenter & Kennedy, 1994), which uses five stages to achieve fourth-order accuracy. The linear terms were treated implicitly with a Crank–Nicolson scheme (Canuto et al., 2007, Appendix D.3), as implemented in JAX-CFD (Kochkov et al., 2021). The step size was constrained by the CFL condition (Courant et al., 1928; Gottlieb & Tadmor, 1991). Specifically,

$$\Delta t \leq \frac{C\Delta x}{v_{\max}},$$

with $v_{\max} = 7$, $C = 0.5$, and $\Delta x = 2\pi/256$, yielding an upper bound of $\Delta t \approx 1.75 \times 10^{-3}$. Since we treat the diffusion implicitly, the upper bound is not related to the viscosity $\nu$.

**Initial condition.** To generate ground-truth states, we first sampled a divergence-free random velocity field, filtered spectrally with a peak wavenumber of 4 and maximum velocity of 7.[9] The system was then integrated forward to $T = 10$ to reach a statistically stationary regime, and the velocity field at $T = 10$ was taken as the initial condition.

### A.4 SANITY CHECK

Throughout our numerical experiments, VANILLA-4DVAR performed poorly once the observation sparsity exceeded 2. Similar limitations have been observed in prior work; for example, Frerix et al. (2021, Figure 9) presents the same issue even when employing background covariance-based preconditioning, suggesting that this challenge is not unique to our setting. Notably, both their implementation and ours are built on JAX-CFD (Kochkov et al., 2021). To ensure correctness, we validated our numerical solver and VANILLA-4DVAR implementation through two sets of tests: convergence tests and assimilation tests.

**Convergence tests.** Figure 5 summarizes the convergence behavior of our numerical solver using log–log error plots. Spatial convergence was tested at a fixed time step $\Delta t = 10^{-4}$, with grid spacings $\Delta x \in \{2\pi/2^8, \ldots, 2\pi/2^4\}$. The results show exponential convergence in space. Temporal convergence was evaluated at a fixed spatial resolution ($\Delta x = 2\pi/2^8$), with time steps $\Delta t \in \{10^{-4}, 2 \times 10^{-4}, \ldots, 2^4 \times 10^{-3}\}$. Here, the observed algebraic rate matches the expected order of the time-stepping scheme.

**Assimilation tests.** To validate the assimilation setup in a simplified regime, we considered a short window $t \in [0, 10^{-2}]$ with dense observations (sparsity level $2^2$). In this case, optimization with L-BFGS successfully converged to the true initial condition, achieving a relative error of $2.64 \times 10^{-5}$ (Figure 6). However, as the observation sparsity increased beyond 3, optimization frequently became trapped in local minima, leading to errors on the order of $10^{-3}$ or higher. We have also considered a non-incremental form (eq. (4)), but the results remained the same.

### A.5 PHYSICS-INFORMED NEURAL NETWORKS

**Physics-informed neural networks.** Consider a generic PDE,

$$\mathcal{D}[u](\xi) = f(\xi), \quad \xi \in \Omega \subset \mathbb{R}^d,$$
$$\mathcal{B}[u](\xi) = g(\xi), \quad \xi \in \partial\Omega,$$

---

[9]See `src.cfd.KolmogorovFlow.initialize_state` in our codebase.

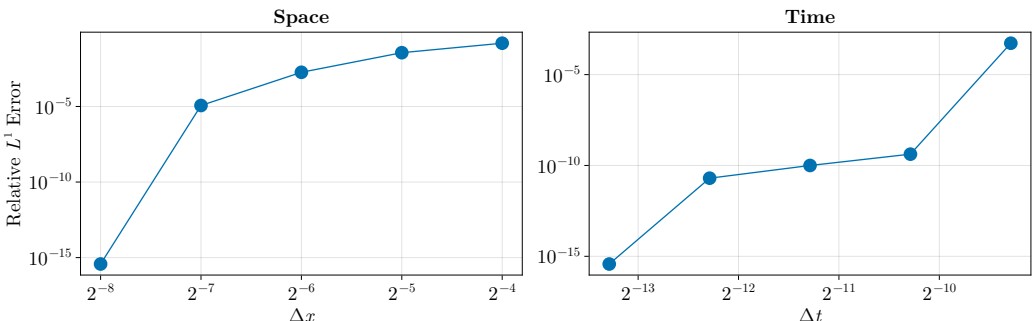

Figure 5: Convergence tests of the numerical solver. Log–log plots of relative $L^1$ error versus grid spacing. The left panel displays exponential spatial convergence at a fixed time step size $\Delta t = 10^{-4}$. The right panel presents an algebraic order of temporal convergence at the fixed spatial resolution $\Delta x = 2\pi/2^8$.

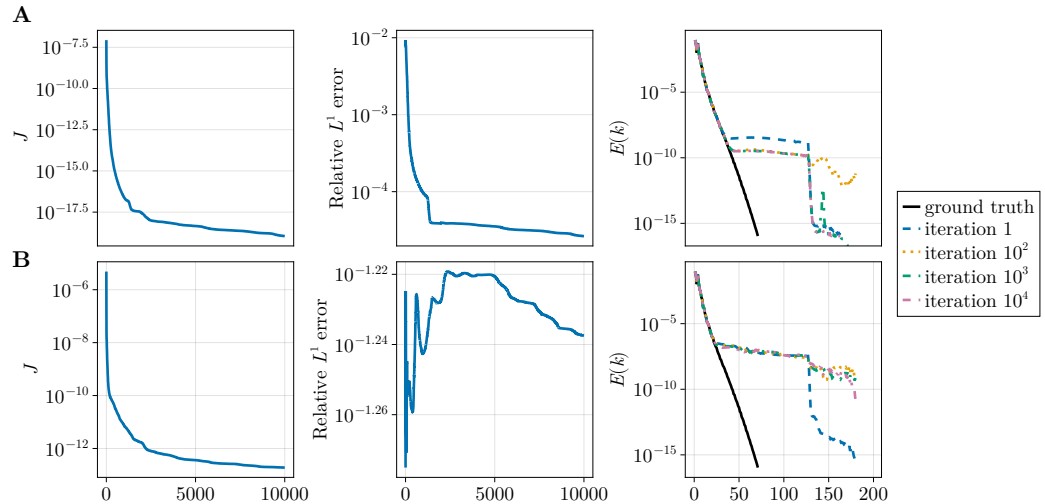

Figure 6: Assimilation test results with VANILLA-4DVAR. Left: cost function decay under L-BFGS optimization. Center: reduction of relative $L^1$ error, confirming consistency between cost decrease and state recovery. Right: energy spectra of assimilated and ground-truth initial conditions. Results are shown for two observation sparsity levels: (**A**) $2^2$, where convergence is successful, and (**B**) $4^2$, where the method begins to diverge.

where $\mathcal{D}$ is a differential operator and $\mathcal{B}$ encodes boundary or initial conditions. Physics-informed neural networks (PINNs) approximate the solution $u$ with a neural network $u^\theta$ and determine the parameters $\theta$ by minimizing

$$L(\theta) = \int_\Omega \left|\mathcal{D}[u^\theta](\xi) - f(\xi)\right|^2 \, \mathrm{d}\xi + \lambda \int_{\partial\Omega} \left|\mathcal{B}[u^\theta](\xi) - g(\xi)\right|^2 \, \mathrm{d}\sigma(\xi),$$

where $\lambda > 0$ balances the PDE residual and boundary conditions. In practice, these integrals are approximated by the Monte Carlo method, and automatic differentiation computes PDE residuals and gradients to $\theta$. reinforcing the potential of PINNs as a general framework for solving inverse problems and PDEs. The idea was originally explored by Lagaris et al. (1998) and popularized by Raissi et al. (2019).

**Separable physics-informed neural networks.** For a function $f : \mathbb{R}^d \to \mathbb{R}$, we approximate it using a separable physics-informed neural network (SPINN) of the form

$$f_{\text{SPINN}}(x_1, \ldots, x_d; \theta_1, \ldots, \theta_d) = \sum_{r=1}^{R} \prod_{i=1}^{d} \text{MLP}_r(x_i; \theta_i),$$

where each $\text{MLP} : \mathbb{R} \to \mathbb{R}^R$ is parameterized by $\theta_i$, and $\text{MLP}_r$ denotes its $r$-th output. This separable structure reduces the computational cost of evaluating tensor-product grids: instead of $O\left(\prod_{i=1}^{d} N_i\right)$ forward passes, only $O\left(\sum_{i=1}^{d} N_i\right)$ evaluations are needed when the $i$-th axis has $N_i$ points. For further details, we refer the reader to Cho et al. (2024).

For numerical experiments illustrated in Section 3, we adopt width 64, depth 3, rank ($R$) 128, and 5 Fourier features. Weights are initialized with Glorot initialization (Glorot & Bengio, 2010).

**Velocity parameterization.** We parameterize the velocity field directly and obtain the vorticity via the curl operator using automatic differentiation. Because this choice does not automatically satisfy the incompressibility condition, we include a divergence-penalty term $\|\nabla \cdot \mathbf{u}\|^2$ in the PINN loss with weight $5 \times 10^3$. Several alternatives are possible. For example, the method of Richter-Powell et al. (2022) enforces incompressibility by construction and avoids the explicit penalty. Parameterizing the vorticity is another option, but it requires solving a Poisson equation on a structured grid, which sacrifices the mesh-free character of PINNs. Likewise, parameterizing the stream function $\psi$ is feasible, though it introduces higher-order derivatives and is therefore less efficient.

## A.6 HYPERPARAMETERS

**VANILLA-4DVAR.** Optimization is performed with L-BFGS (Liu & Nocedal, 1989) using a history size of 10. The increment $\delta u_0$ is initialized as zero. Training is run for 1K steps.

**NEURAL-4DVAR and HYBRID-4DVAR.** Optimization is performed with AdamW (Loshchilov & Hutter, 2019) using an initial learning rate of $10^{-2}$, cosine learning-rate decay (Loshchilov & Hutter, 2017), and 1K training steps. The increment $\delta u_0$ is parameterized by a SPINN.

**PINN-4DVAR.** Optimization is performed with AdamW using an initial learning rate of $10^{-3}$, cosine learning-rate decay, and 10K training steps. The full spatio-temporal state $u(t, x)$ is parameterized directly with a SPINN. For $L_{\text{PINN}}(\theta)$, we employ $N_t = N_x = N_y = 128$ collocation points, sampled randomly from the uniform distribution—$\text{Uniform}[0, 0.5]$ for time and $\text{Uniform}[0, 2\pi]$ for space in every iteration. For a noise level $\sigma$, we set $\lambda_{\text{data}} = 1/\sigma^2$. For $\sigma = 0$, we set $\lambda_{\text{data}} = 5 \times 10^3$.

## A.7 HYPERPARAMETER ANALYSIS AND ABLATION STUDY

**Hyperparameter sweep for NEURAL-4DVAR.** We conducted a hyperparameter sweep for NEURAL-4DVAR, varying the network width $\{2^5, 2^6, 2^7\}$, depth $\{2, 3, 4, 5\}$, initial learning rate $\{10^{-2}, 10^{-3}, 10^{-4}\}$, and number of Fourier modes $\{1, 3, 5, 7\}$. All experiments were performed at sparsity level $32^2$ with $10^3$ training epochs. Table 4 reports the results, highlighting a notable mismatch between cost reduction and state accuracy: while the failure case achieves the lowest cost, it yields the highest error, indicating overfitting to spurious high-frequency modes.

**Hyperparameter sweep for PINN-4DVAR** We conducted a hyperparameter sweep for PINN-4DVAR, varying the network width $\{2^4, 2^5, 2^6, 2^7\}$, depth $\{2, 3, 4, 5\}$, initial learning rate $\{10^{-2}, 10^{-3}, 10^{-4}\}$, number of Fourier modes $\{1, 3, 5, 7\}$, and training epochs $\{10^3, 5 \times 10^3, 10^4\}$. Table 5 reports the top five performing settings (ranked in accuracy) along with two failure cases. We observed that large networks tend to diverge when trained with an initial learning rate of $10^{-2}$ (see Failure cases 1 and 2 in the Table 5). By contrast, using smaller learning rates ($\leq 10^{-3}$) consistently stabilized training, yielding relative $L^1$ errors below 0.5. Surprisingly, a small SPINN of width 16, depth 2, and 5 Fourier modes was sufficient to reach the relative error less than 0.4.

| | | HYPERPARAMETERS | | | RESULTS | |
|---|---|---|---|---|---|---|
| | width | depth | $\text{lr}_0$ | Fourier modes | error | cost |
| Top 5 | 128 | 5 | $10^{-2}$ | 3 | **3.519E-1** | 5.962E-6 |
| | 128 | 4 | $10^{-2}$ | 3 | 3.749E-1 | 4.884E-6 |
| | 128 | 5 | $10^{-3}$ | 3 | 3.901E-1 | 2.728E-5 |
| | 64 | 5 | $10^{-2}$ | 3 | 4.042E-1 | 6.920E-6 |
| | 64 | 5 | $10^{-2}$ | 3 | 4.100E-1 | 7.413E-6 |
| Failure case | 64 | 5 | $10^{-2}$ | 7 | 9.163E-1 | **4.346E-6** |

Table 4: Hyperparameter sweep results for NEURAL-4DVAR. Boldface and underline indicate the best and the second best in each column. The top five configurations (ranked by accuracy) and the failure case show that lower error does not necessarily coincide with lower cost.

| | | HYPERPARAMETERS | | | | RESULTS | |
|---|---|---|---|---|---|---|---|
| | width | depth | $\text{lr}_0$ | Fourier modes | epochs | error | runtime (s) |
| Top 5 | 16 | 2 | $10^{-2}$ | 5 | $10^4$ | 3.641E-1 | 42 |
| | 32 | 2 | $10^{-3}$ | 7 | $10^4$ | 3.606E-1 | 43 |
| | 32 | 3 | $10^{-3}$ | 7 | $10^4$ | 3.605E-1 | 93 |
| | 32 | 4 | $10^{-3}$ | 3 | $10^4$ | 3.585E-1 | 103 |
| | 32 | 5 | $10^{-3}$ | 5 | $10^4$ | **3.578E-1** | 128 |
| Failure case 1 | 128 | 3 | $10^{-2}$ | 7 | $5 \times 10^3$ | 3.864E-1 | 66 |
| | | | | | $10^4$ | 8.146E+2 | 89 |
| Failure case 2 | 128 | 4 | $10^{-2}$ | 5 | $5 \times 10^3$ | 3.826E-1 | 96 |
| | | | | | $10^4$ | 1.398E+2 | 121 |

Table 5: Hyperparameter sweep results for PINN-4DVAR. The failure cases indicate that too large an initial learning rate $10^{-2}$ causes divergence after $5 \times 10^3$ steps.

## B   ADDITIONAL FIGURES

Figures 7 to 10 illustrate several aspects of the assimilation process. Panel (**A**) shows the ground-truth initial vorticity. Panel (**B**) tracks the evolution of both the 4DVAR cost function and the relative $L^1$ error between the estimated and true initial conditions over optimization steps. Panel (**C**) presents the corresponding energy spectrum of the assimilated state, while Panel (**D**) visualizes the progression of the estimated initial vorticity at selected optimization steps.

## C   THE USE OF LARGE LANGUAGE MODELS

We have used large language models (OpenAI, 2025) to polish and revise texts.

(a) Regression results.

|  | | Sparsity $k^2$ | | | | |
| --- | --- | --- | --- | --- | --- | --- |
|  | | $2^2$ | $4^2$ | $8^2$ | $16^2$ | $32^2$ |
| | 0% | **4.16E-2** | **4.17E-2** | **8.26E-2** | 2.97E-1 | 8.50E-1 |
| | 5% | **5.30E-2** | **1.09E-1** | 2.70E-1 | 3.61E-1 | 8.51E-1 |
| $\sigma$ | 10% | **1.00E-1** | 4.40E-1 | 9.00E-1 | 4.80E-1 | 8.61E-1 |
| | 15% | **2.25E-1** | 1.01E0 | 1.24E0 | 4.68E-1 | 8.59E-1 |
| | 20% | 4.36E-1 | 1.62E0 | 1.76E0 | 6.16E-1 | 8.76E-1 |
| | 25% | 7.61E-1 | 2.18E0 | 2.04E0 | 6.78E-1 | 9.07E-1 |

(b) PINN results.

|  | | SPARSITY | | | | |
| --- | --- | --- | --- | --- | --- | --- |
|  | | $2^2$ | $4^2$ | $8^2$ | $16^2$ | $32^2$ |
| | 0% | 8.54E-2 | 8.54E-2 | 8.75E-2 | **1.56E-1** | **3.62E-1** |
| | 5% | 2.24E-1 | 2.24E-1 | **2.24E-1** | **2.52E-1** | **3.82E-1** |
| $\sigma$ | 10% | 3.13E-1 | **3.12E-1** | **3.13E-1** | **3.27E-1** | **4.08E-1** |
| | 15% | 3.60E-1 | **3.60E-1** | **3.63E-1** | **3.67E-1** | **4.44E-1** |
| | 20% | **3.94E-1** | **3.92E-1** | **3.95E-1** | **3.99E-1** | **4.74E-1** |
| | 25% | **4.26E-1** | **4.24E-1** | **4.27E-1** | **4.32E-1** | **4.97E-1** |

Table 6: Ablation study evaluating the effect of the physics-informed loss. Rows correspond to noise levels and columns to sparsity levels. (a) Optimization without the physics-informed loss, minimizing only observation misfits. (b) Optimization with the physics-informed loss included. Bold entries indicate higher accuracy between (a) and (b). While regression without physics constraints performs well under low noise and dense observations, PINN-4DVAR consistently outperforms it as noise and sparsity increase.

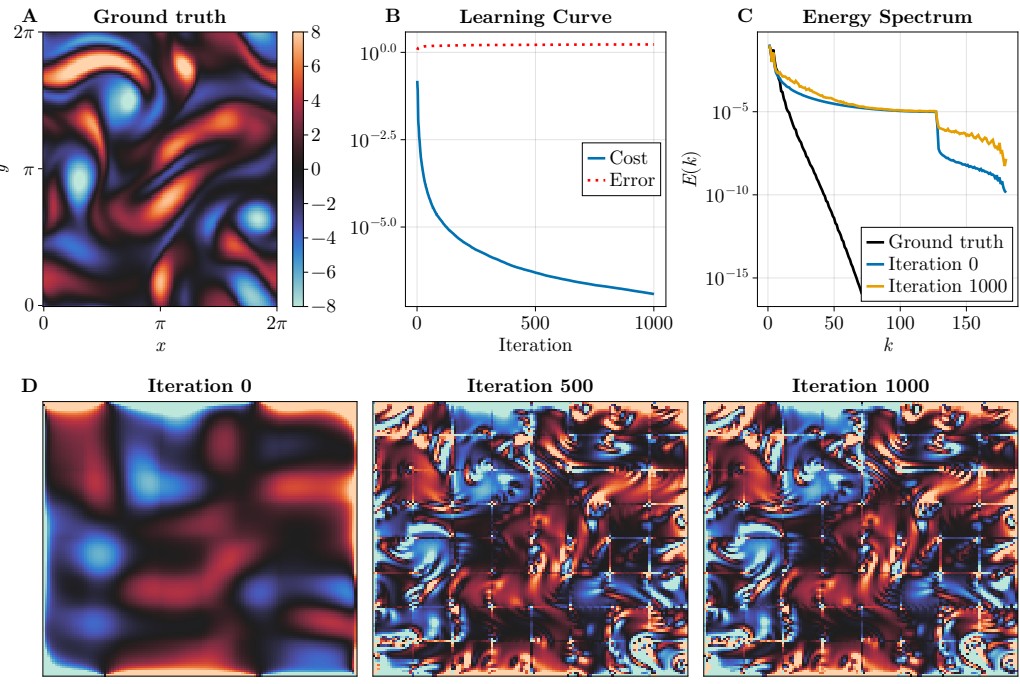

Figure 7: VANILLA-4DVAR. (**B**) While the cost function decreases, the error plateaus. (**C**) Compared to the energy spectrum of the initial guess (Iteration 0), the energy spectrum of the optimized initial condition (Iteration 1000) has higher energy in high-wavenumbers.

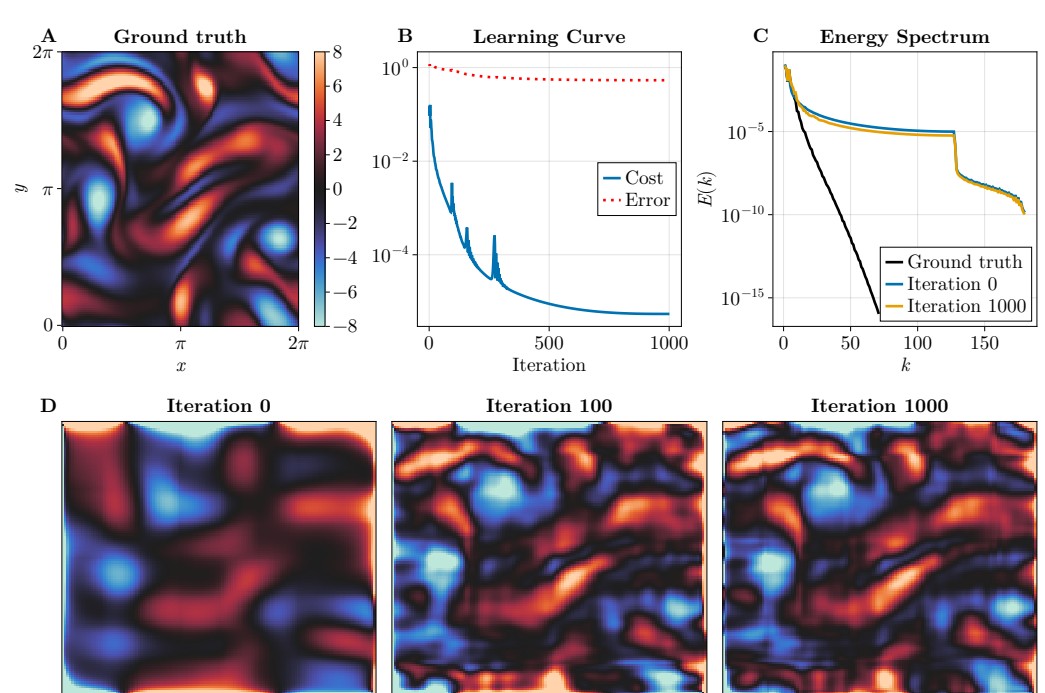

Figure 8: NEURAL-4DVAR.

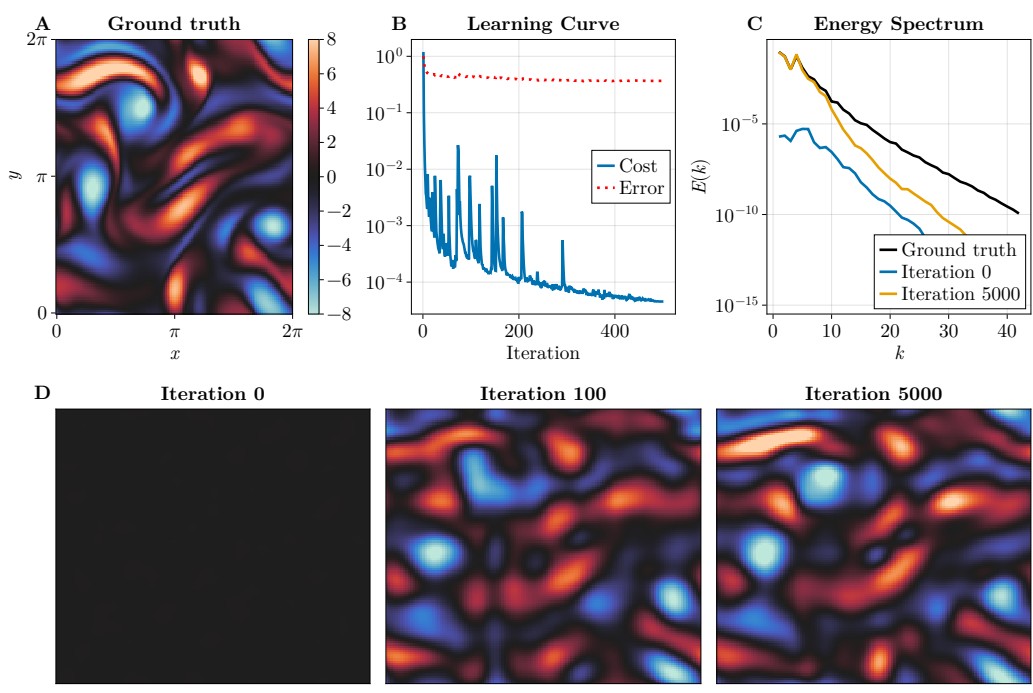

Figure 9: PINN-4DVAR.

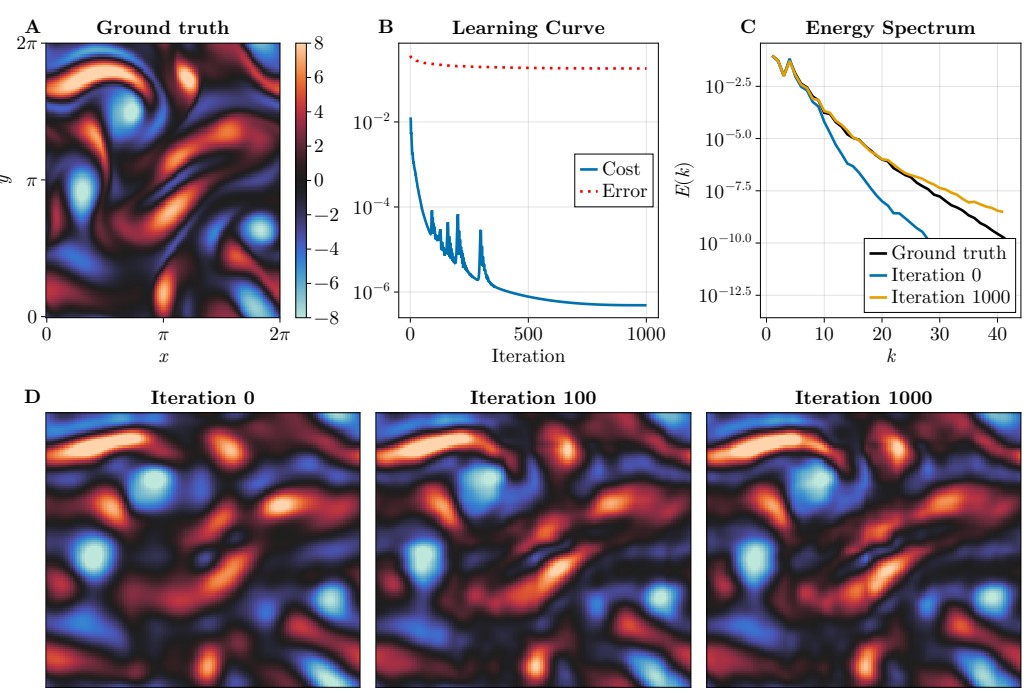

Figure 10: HYBRID-4DVAR.

