# OpenReview forum: "Reparameterizing 4DVAR with neural fields"
_ICLR.cc/2026/Conference — ICLR 2026 Conference Withdrawn Submission_

### Official Review · Reviewer_mMTv · 2025-10-23

**Soundness:** 1
**Presentation:** 3
**Contribution:** 1
**Rating:** 2
**Confidence:** 4

**Summary:**

The paper presents a genuinely innovative integration of neural fields with 4DVar to enable parallel data assimilation, showing promising efficiency and accuracy on 2D Kolmogorov flow. However, the central motivation that 4DVar cannot be parallelized overlooks weak-constraint 4DVar, whose formulation closely resembles the proposed PINN-4DVar and can be parallelized. Important implementation details (e.g., how the PINN integral is computed) are missing, and the Vanilla 4DVar baseline omits the background-error term, likely causing the reported spurious high-frequency artifacts. Most critically, the manuscript lacks a thorough literature review and strong baselines, especially comparisons to weak-constraint 4DVar and to a conventional hybrid pipeline (weak- then strong-constraint), which substantially weakens the contribution. Strengthening the related work and adding these baselines would better substantiate the method’s advantages in both accuracy and efficiency.

**Strengths:**

The paper tackles an important data assimilation problem. To address the challenge of parallelizing 4DVar, it introduces a novel approach that represents the full field with a neural field, enabling parallel optimization. To my knowledge, this is the first attempt to integrate neural fields with 4DVar. Experiments on the 2D Kolmogorov flow model show measurable gains in both efficiency and accuracy over a conventional 4DVar baseline. The presentation is also very clear.

**Weaknesses:**

- The paper is motivated by the claim that traditional 4DVar cannot be parallelized, and therefore proposes a parallelizable 4DVar assimilation algorithm. However, this claim is not accurate. The “4DVar” referred to in the paper is the strong-constraint 4DVar. There is also weak-constraint 4DVar [1] [2], which does allow parallel optimization of the state at different time levels, though it cannot strictly enforce temporal physical consistency—this is imposed only weakly via a penalty.

  The formulation of weak-constraint 4DVar is very similar to the proposed PINN-4DVar. Its cost function (using the paper’s notation) is:
  $$
  L(u)=\int\_\Omega\sum\_{i=0}^{T/\Delta t-1}||\mathcal{M}\left(u(i\Delta t,x)\right)-u((i+1)\Delta t, x)||\_Q^2\text{d}x + \lambda\_{data}\sum\_{k=0}^K\left(H(u_k)-y_k\right)^2
  $$
  This cost directly optimizes the entire sequence of fields. The first term represents the dynamical loss along the trajectory; $\mathcal{M}$ is the operator that advances the field by $\Delta t$ (i.e., the time-$\Delta t$ solution operator of the PDE in the paper). This is essentially the same as the PINN loss used in the manuscript; the only difference is that weak-constraint 4DVar does not represent the full field as a neural field. Because weak-constraint 4DVar can also be parallelized, the paper’s central claim is substantially weakened. The authors should clearly discuss the advantages of PINN-4DVar over weak-constraint 4DVar and include experimental comparisons.

- In addition, when discussing the drawbacks of Vanilla 4DVar, the paper attributes “spurious high-frequency perturbations” to the method. This seems to stem from the implementation omitting the background-error term $||u_0-u_b||_B^2$. In a complete Vanilla 4DVar, the $B$ matrix constrains the analysis increment $u_0 - u_b$ to a smoother subspace [3] [4]. Spurious high-frequency components would inflate the background term and therefore are strongly penalized during assimilation, making them unlikely to appear. Consequently, the claimed advantage attributed to the neural field on this point does not hold.

[1] Fisher, Mike, et al. "Weak-constraint and long-window 4DVar." *ECMWF Technical Memoranda* 655 (2011): 47.

[2] Fablet, Ronan, et al. "Learning variational data assimilation models and solvers." Journal of Advances in Modeling Earth Systems 13.10 (2021): e2021MS002572.

[3] Bannister, Ross N. "A review of forecast error covariance statistics in atmospheric variational data assimilation. I: Characteristics and measurements of forecast error covariances." *Quarterly Journal of the Royal Meteorological Society: A journal of the atmospheric sciences, applied meteorology and physical oceanography* 134.637 (2008): 1951-1970.

[4] Bannister, Ross N. "A review of forecast error covariance statistics in atmospheric variational data assimilation. II: Modelling the forecast error covariance statistics." *Quarterly Journal of the Royal Meteorological Society: A journal of the atmospheric sciences, applied meteorology and physical oceanography* 134.637 (2008): 1971-1996.

**Questions:**

- How is the integral for the PINN term actually computed in the paper? The implementation details don’t seem to be specified.
- Compared with weak-constraint 4DVar, does the proposed PINN-4DVar retain an advantage in computational efficiency and accuracy?
- By analogy with the paper’s Hybrid-4DVar, a conventional workflow could first run weak-constraint 4DVar for an initial solution and then refine it with strong-constraint 4DVar. Relative to this conventional Hybrid-4DVar, does the proposed Hybrid-4DVar still offer clear benefits?

---

### Official Review · Reviewer_XNhP · 2025-10-27

**Soundness:** 2
**Presentation:** 2
**Contribution:** 2
**Rating:** 2
**Confidence:** 4

**Summary:**

## Summary
This paper proposes a novel reformulation of four-dimensional variational data assimilation (4DVAR) by reparameterizing the problem using neural fields. The core idea is to represent the spatiotemporal state as a continuous function parameterized by a neural network. The authors introduce two main variants: NEURAL-4DVAR, which parameterizes only the initial condition $u_0^\theta$, and PINN-4DVAR, which parameterizes the spatiotemporal state $u_0^\theta(t,x)$ and enforces physical constraints via a physics-informed loss. A major advantage of this framework over many machine learning-based approaches is its independence from large-scale datasets; it does not require access to ground-truth states or reanalysis data. Experiments on the 2D incompressible Navier-Stokes equations demonstrate these benefits.

**Strengths:**

## Strengths
- Unlike many machine learning approaches for DA, this paper introduce a framework that does not need to be pre-trained on large ground-truth or reanalysis datasets. The neural fields are optimized from scratch for each assimilation window using only the available observations
- The PINN-4DVAR formulation breaks the time-sequential dependency of classical 4DVAR. This enables parallel-in-time optimization, which resulted in a runtime speedup. Crucially, the method retains the core principle of 4D-Var by enforcing the time-evolutionary dynamics across the entire window as a soft constraint via the PINN loss.

**Weaknesses:**

## Weakness
- The paper explicitly omits the background error covariance $B$ term, which is important in operational 4DVAR for regularization. The authors claim the neural field's *spectral bias* acts as an implicit regularizer to prevent non-smooth solutions but this requires clearer justification and evidence. Empirically, it is unclear how the proposed method would compare to a properly regularized baseline.
- The paper's validation is limited to a single benchmark (2D Kolmogorov flow). Including an additional benchmark with different physical properties, such as the shallow water equations, would be necessary to substantiate the claims of empirical performance and generality.
- The paper’s best results come from HYBRID-4DVAR (PINN initialization + NEURAL-4DVAR refinement). This implies neither core method suffices alone: PINN-4DVAR is faster but weaker in rollout accuracy, while NEURAL-4DVAR improves accuracy but inherits the sequential bottleneck. The hybrid reads as an adhoc combination rather than a unified framework.

**Questions:**

## Questions
- Could the authors please elaborate on the model architecture? Appendix A.5 mentions the use of "5 Fourier features" for the SPINN. How are these implemented?
- The experimental setup uses L-BFGS for VANILLA-4DVAR but AdamW for the NEURAL- and PINN-4DVAR variants. While this is a common choice, could the authors comment on how this choice might affect the performance comparison? Is there memory limitation to use L-BFGS for NEURAL-4DVAR?
- Could authors comment on the spurious articfacts in the assimiated initial state $u_0$ for nearly all the methods, shown in Figures 2, 7, 8, 9, and 10.
- Since the introduced PINN loss is estimated via Monte-Carlo over the full space–time domain, Could the author elabroate the impact of discretization: (i) time-step and integrator choice, and (ii) spatial resolution.
- The PINN-4DVAR method assumes the governing equations $\mathcal{F}$ are perfectly known. How robust is the method to model error (e.g., from unresolved physics, discretization differences, or uncalibrated parameters), which is a common challenge in operational forecasting?

---

### Official Review · Reviewer_fWtC · 2025-10-30

**Soundness:** 3
**Presentation:** 3
**Contribution:** 2
**Rating:** 4
**Confidence:** 4

**Summary:**

This paper reformulates classical 4D variational data assimilation using neural fields, allowing the state to be represented as a continuous neural function and optimized in parallel over time.  While results on 2D Navier–Stokes show moderate accuracy and efficiency gains, the PINN constraint remains weak and does not guarantee true physical consistency.

**Strengths:**

The paper presents a well-motivated reformulation of 4D variational data assimilation using neural fields, bridging classical optimization-based approaches and implicit neural representations. Its main strength lies in demonstrating that neural parameterization can stabilize optimization and reduce computational cost through parallel-in-time training, leveraging the spectral bias of neural fields to suppress high-frequency artifacts. The methodology is clearly described, theoretically sound at the conceptual level, and supported by thorough ablation studies, runtime analysis, and spectral diagnostics.

**Weaknesses:**

I have serious doubt about involving PINNs loss since what if the reality does not satisfied with the PDE closure modelling like weather forcast. Some assumption is even wrong from the human knowledge.

**Questions:**

1. Please explain what happened if there's no garantueen equation based modelling for your system.
2. some small scale frequecies may be neglected by fourier truncation. how do you deal with it?

---

> ### Author Response · Authors · 2025-11-12
>
> Dear reviewer fWtC,
>
> We appreciate your time and effort in reviewing this article. We genuinely hope that the subsequent discussion will be beneficial for both of us.
>
> **Weaknesses**
> > I have serious doubts about involving PINNs' loss since what if the reality does not satisfy the PDE closure modelling, like a weather forecast. Some assumptions are even wrong from human knowledge.
>
> Yes, we also agree on this point. Current numerical weather prediction is based on primitive equations. Of course, the model has many assumptions, since resolving all small-scale dynamics has been impossible. However, scientists have tried their best to devise models that can simulate nature. Such models are devised based on scientists' observations, and then scientists "summarize" their observations in compact forms. We would like to say that, rather than depending solely on data, incorporating imperfect models into the process would be better in multiple ways.
>
>
> **Questions**
> > Please explain what happened if there's no guaranteed equation-based modelling for your system.
>
> If no information is known, then the most straightforward approach is to minimize observation misfit
>
> $$
> J(u_0) = \| y_0 - H(u_0) \|_2^2.
> $$
>
> If partial information is known, then there could be multiple ways to tackle the problem. For instance, we may first estimate unknown parameters for differential equations, and then apply the proposed framework. Or, joint optimization might offer a better performance.
>
> > Some small-scale frequencies may be neglected by Fourier truncation. How do you deal with it?
>
> To our knowledge, there is no general way to resolve all small-scale features ($\approx$ high-frequency features), due to the presence of round-off errors in the floating point arithmetic. We considered a sufficient enough resolution $256^2$ to approximate ground truth states (please see Figure 4 panel C for the ground true state's power spectrum). However, we will reach the regime where a compromise is necessary if the Reynolds number and the forcing become higher and stronger, respectively. We left this case as a future work.

---

### Official Review · Reviewer_xyDL · 2025-11-01

**Soundness:** 3
**Presentation:** 2
**Contribution:** 2
**Rating:** 2
**Confidence:** 5

**Summary:**

This paper proposes a neural field-based reformulation of four-dimensional variational data assimilation (4DVAR) for numerical weather prediction. The key innovation is representing the full spatiotemporal state as a continuous function parameterized by a neural network, which enables parallel-in-time optimization and eliminates the time-sequential dependency of classical 4DVAR. The method is evaluated on 2D incompressible Navier-Stokes equations with Kolmogorov forcing and shows promise in producing stable initial condition estimates without requiring ground-truth reference data.

**Strengths:**

- The core idea of using neural fields to regularize 4DVAR is novel and merits further exploration
- The approach removes time-sequential dependencies, enabling parallel optimization
- Unlike typical ML approaches, the method does not require access to ground-truth states or reanalysis data, which broadens its applicability
- The physics-informed loss formulation is a natural way to incorporate physical constraints

**Weaknesses:**

Presentation and clarity issues:
- The model definition lacks clarity. After introducing the discrete state-space formulation in Section 1.1, the paper should maintain this formulation consistently throughout
- The definition of $L_{PINN}$ is unclear—why switch to a continuous formulation here? This loss term requires more precise mathematical specification
- The variable $n$ (page 5, "Simulated observations" section) appears to be undefined
- The notion of "sparsity" and the definition of the observation operator $H$ need explicit clarification

Organization:
- The Kolmogorov flow discussion should be moved to the appendix, with only the discrete formulation presented in the main paper
- Conversely, the neural network architecture details should be moved from the appendix to the main paper

Experimental concerns:
- Figure 4 (panels A and B) suggests VANILLA-4DVAR may require additional regularization, as the loss decreases while L1 error increases. This raises questions about experimental fairness and whether baselines are treated with sufficient care
- To address concerns about fair comparison, the results should be benchmarked against published 4DVAR results from the literature

**Critical omission:** No comparison is provided with weak-constraint 4DVAR, which appears conceptually similar to the proposed PINN-4DVAR approach. This comparison is essential.

Code was not submitted with the paper, which limits reproducibility

**Questions:**

1. For the hybrid approach: Is PINN-4DVAR used solely to provide a good initialization for the NEURAL method, or does it serve additional purposes?
2. How would the proposed method compare quantitatively to weak-constraint 4DVAR formulations?

---

### Note · Authors · 2025-11-14

**Comment:**

Dear reviewers, I have decided to withdraw the manuscript. Thank you very much for taking your time to review this manuscript.

**Withdrawal Confirmation:**

I have read and agree with the venue's withdrawal policy on behalf of myself and my co-authors.